# The Income Elasticities of Food, Calories, and Nutrients in China: A Meta-Analysis

**DOI:** 10.3390/nu14224711

**Published:** 2022-11-08

**Authors:** Jinlu Zhao, Jiaqi Huang, Fengying Nie

**Affiliations:** 1Agricultural Information Institute of Chinese Academy of Agricultural Sciences, Beijing 100081, China; 2College of Economics and Management, China Agricultural University, Beijing 100083, China

**Keywords:** food consumption, nutrient, calories, income elasticity, meta-analysis

## Abstract

Estimating food- and nutrient-income elasticities is important for making food and nutrition policies to combat malnutrition. There are many studies that have estimated the relationship between income growth and food/nutrient demand in China, but the results are highly heterogeneous. We conducted a meta-analysis in China to systematically review the elasticity of food, calories, and other nutrients to income. We considered a meta-sample using a collection of 64 primary studies covering 1537 food-income elasticities, 153 nutrient-income elasticities, and 147 calorie-income elasticity estimates. There are significant differences in the size of the income elasticities across food and nutrient groups. We found that food- and calorie-income elasticity appear to decline as per capita income increases, except for vitamin and aquatic products. We also found a publication bias for food and calories, and in particular, the study attributes may be important, as they can influence estimates. Given the limited study on nutrient-income elasticity, understanding the impact of income changes on nutrient intake is an important direction worthy of further research.

## 1. Introduction

Ending poor diets and malnutrition in all its forms is a goal that is intrinsically linked with some of the world’s most pressing challenges. The most recent data show that an unacceptably large number of people are still affected by malnutrition. Globally, of all children under five years of age, one in five is stunted (149.2 million), 45.4 million (6.7%) are wasted, and 38.9 million (5.7%) are overweight [1]. Nutritional burdens trigger some chronic non-communicable diseases and cause enormous health and economic loss in society. Poor diet is the main contributor to nutritional burdens. It is estimated that nearly 690 million people suffer from chronic malnutrition, 3 billion people cannot afford a healthy diet, and 11 million die each year from low-quality diets worldwide [2]. Given the consequences of poor diets, it is particularly important to design evidence-based policies to encourage a healthy diet. Studying people’s food and nutrient demand is essential for policymaking because it explains how economic and policy changes influence people’s demand for food and nutrients.

The coexistence of malnutrition and “hidden hunger” is a common challenge for many emerging economies, including China [3,4]. As the largest developing country, China’s population accounts for about one-fifth of the world [5]. Following the rapid economic growth, the per capita disposable income of Chinese residents saw an average annual growth of 7.2%, in real terms, after adjusting for inflation, from 2011 to 2020. Meanwhile, China’s food consumption pattern has changed from supply-constrained to nutrition-oriented [6]. According to official statistics, China’s share of undernourished people has declined substantially (from 10.6% in 2000–2002 to below 2.5% in 2017–2019) [7]. Yet, there are at least 250 million “hidden hungry (hidden hunger refers to the hunger symptoms of hidden nutritional needs caused by the body’s nutritional imbalance or lack of certain vitamins and essential minerals)” people in China [1], which is characterized by micronutrient deficiency, which is a cause for concern. At the same time, China now finds itself among the nations greatly affected by the growing population of overweight or obese individuals, with an estimated one in six Chinese adults falling into these BMI classifications and childhood and adolescent obesity rates rising over the past several decades [8]. About 70% of chronic non-communicable diseases are linked to hidden hunger and malnutrition [7]. Particularly, the mortality of chronic non-communicable diseases accounted for 85% of all deaths in 2021 and has become a serious concern in China, according to the Report on Chinese Residents’ Chronic Diseases and Nutrition [9]. Given the consequences of poor diet, the size of China’s population, and the rapid growth of per capita income, it is important to study the heterogeneity of Chinese food and nutrient demand with changes in income, providing an evidential basis for designing policies to improve nutrition.

Generally, food demand is income inelastic (elasticities are less than one), reflecting Engel’s law that food budget shares decline when income rises [10]. (Income elasticities of food demand are economic measurements of the responsiveness of food consumption to income for a group of consumers. The income elasticity of food demand measures the percent change in the consumption of total food or a certain food item or group of food items to a percent change in the real income of consumers [11]). However, a series of articles emerged, casting doubt on the role of income. The role of income continues to be interesting, since contrasting results appear throughout the literature [12,13,14]. The food demand–income elasticities estimated in the literature in China showed heterogeneous results. The food demand–income elasticities estimated in the literature ranged from −1.3 to 2.3 (see Figure 1). The income elasticity for staple food reached over 1.0 in a rural study by Han et al. [15]. and Hovhannisyan et al. [16] found that for urban people, the income elasticity for staple food was close to 0, and Carter and Zhong [17] found a negative value. Most studies estimated income elasticities for meat are higher than 0, and some found greater than 1 [6,18,19]. The income elasticities of calories range from −0.4 to 0.9 (see Figure 2). Several authors, such as Liu and Hu [20], Tian [21], and Zhang [22], reported a strong relationship between the level of per capita expenditure and calorie intake. In contrast, Nie and Poza [23] and Chen [24] found that the relationship between household income and calorie intake is not significantly different from zero. Tian and Yu [25] found a nonlinear relationship between nutrition indices and income growth. Households with moderate or higher macronutrient intake tend to decrease their macronutrient intake—especially fat—in rural areas, while in urban areas, the least nourished tend to consume more carbohydrates and proteins [26]. The results are divergent due to different samples, study regions, methods, and other factors.

This study emphasizes the heterogeneity of food-, calorie- and nutrient-income elasticities in China through a meta-analysis approach and builds on previous studies in three ways. First, we analyze a comprehensive set of food and nutrient products at a disaggregate level. Most quantitative reviews of food demand elasticities focused only on calories [27,28] or only on food [4,27,28,29,30,31,32,33,34,35,36,37,38]. Most of these previous meta-analyses focused on a small number of products or products at a more aggregate level than our study. Second, to the best of our knowledge, this is the first review in China that examines the estimates for income elasticity of food, calorie, and nutrients on a comparative basis. Only three meta-analysis studies of food/nutrient demand were published for China: two are on food products [32,39], the other is on calories [29], and none tackles nutrients. The previous meta-analyses of food and nutrient demand exist for other countries or across countries [11,38,40] but not for China, a country with the largest population experiencing a nutrition transition. 

Third, we consider a comprehensive set of potential factors affecting changes in income elasticity, which is linked to the attributes of primary studies. This study updates the estimates of food- and nutrients–income elasticities in China by a systematic approach and provides evidence that may help design food policies in China.

The rest of the paper is structured as follows: Section 2 introduces the data and methods used on income elasticities for food, calorie, and nutrients in China; Section 3 elaborates on the key descriptive statistics of the meta-sample; Section 4 presents an overview of the meta-analysis and model specification estimated in this paper; and Section 5 reports the results, while the discussion section concludes.

## 2. Description of Data

### 2.1. Selection of Primary Studies and Construction of Meta-Sampler

The analysis started with a literature review retrieval to collect food- and nutrient-income elasticities from relevant primary studies. To locate candidate studies, we conducted the initial search across various online databases, including both published literature (journal articles) and “gray” literature (working papers/reports), using the keywords of “China,” “food,” “calorie,” “nutrient,” “demand,” “consumption,” and “income elasticity.” The databases searched were the following: Google, Google Scholar, Web of Science, EconLit, Scopus, the China National Knowledge Infrastructure (CNKI), FAO, and the World Bank. In addition, we conducted an exhaustive search in the references of review studies of China’s food/nutrient demand e.g., [12,32,37,38,39], and we identified 13 new records. 

We based the selection of food-income elasticities on the following criteria. First, after reading and retrieving articles based on the relevance of the abstract to the research objectives, we excluded review articles and repeated literature. This process eventually yielded 898 comparable studies using China data. Second, the primary criteria used in selecting studies for the current analysis was the presence of standard errors or *t*-values for the computed elasticities, and articles did not fulfill this criterion were excluded, reducing the number of articles to 297. Third, to avoid problems of comparability between income-elasticity estimates, we only maintained studies providing unit-free elasticity estimates of food demand concerning income, further reducing the final sample to 71 studies. Fourth, in the case of multiple model estimates for the same dataset, we included only the authors’ recommended model if our determinants do not capture model differences. Finally, we deleted additional observations with food-, calorie-, and nutrient-income elasticities exceeding five standard deviations outside the means to avoid extreme values, resulting in the removal of 7 observations (see Appendix A for a complete presentation of the data and sources). Our final dataset consisted of 64 articles that matched our objective. The study employed primary interest data comprising the dependent variables (meta variables) (income elasticities for food, calories, and nutrients). The resulting samples without outliers provided 1537 food-income elasticity estimates, 153 nutrient-income elasticity estimates, and 147 calorie-income elasticity estimates (where a study produced multiple income elasticities (e.g., for different food/nutrient groups, for urban and rural samples separately, using different estimation models), all estimates were included in the meta-sample, resulting in a total of 147–1537 elasticities) (see Figure 3).

### 2.2. Descriptive Analysis

The explanatory factors can be divided into two categories—contextual and methodological factors—to explain the heterogeneity in income elasticity estimates [41]. The contextual difference may be generated by food or nutrient categories, locations studied, and time periods. (Some meta-analyses use the time period of the research data as an explanatory variable. The reason why we ‘do not use this variable is that it is highly correlated with the explanatory variables in our model (especially the logarithm of per capita income).) Methodological factors may include study design and budgeting, demand models, estimation procedures, and peer-review [36].

Table 1 summarizes the main characteristics of three categories of research samples: food-, calorie-, and nutrient-income elasticities. Table 1 presents the definition of the variables included in this research. Among the three types of research samples, the number of foodstuffs accounted for about 84% of the observations; the number of elasticities attributed to nutrients and calorie contributions was less observed, at about 8% each. Overall, the mean of the income elasticities was highest for food (0.690), with a standard deviation of 0.685, which was followed by nutrients (0.298), with a standard deviation of 0.315, and calories (0.212), with a standard deviation of 0.225. These statistics indicate large variations in income-elasticity estimates for food, calories, and nutrients warrant further investigation.

#### 2.2.1. Product Differences

The income elasticities vary across different types of food and nutrients. The results show considerable differences across previous studies. The food demand income elasticities estimated in the literature ranged from −1.30 to 2.30 (see Figure 1). The main food grouping in this research is the staple food, vegetables and fruit, meat, oil and fat, aquatic products, dairy products, eggs, and other foods (in general, “staple food” is defined as all cereals, wheat, rice, and coarse grains, and “meat” is defined as all meats (including pork, beef, mutton, and poultry)). In theory, the income elasticity for a food group should be a weighted average of products’ income elasticities, as the collected elasticities are taken from different studies at different times and places. The product classifications are consistent with those in the rural and urban household surveys conducted annually by the National Bureau of Statistics of China (NBSC). (Along with the degree of substitution, that is, foods of the same category should have greater substitutability than those of different categories. The “tubers, starch” are summarized as “staple food”, and “sugar and beverages” is merged into “other food” to facilitate meta-regression analysis, with the limited amount of starch, tubers, sugar, and beverages observed in the primary studies.) Food items that make up basic diets, such as staple food, oil, and fruits and vegetables, have lower income elasticities, with a mean elasticity lower than 0.5. However, elasticities are considerably higher for foods sourced from animals. Food groups with higher elasticities would typically supplement basic diet requirements in China (i.e., aquatic products and dairy products) [42], with elasticities ranging between 1.066 and 1.084.

The income elasticities of calories are −0.40 to 0.90, and nutrients are 0.06 to 1.20 (see Figure 2). The mean nutrient elasticities are high, especially for vitamins and fats (Table 1). Nutrients are divided into macronutrients (protein and fat) and micronutrients (vitamins and minerals). The mean elasticity is the lowest for calories (0.212), which is reasonable given that the Chinese have fulfilled the first-order caloric needs [25]. Foods of lower nutritional value and lower-quality diets generally cost less per calorie and tended to be selected by groups of lower socioeconomic status [10]. Conversely, vitamins, protein, and fat-based products are supplementary to the diets of most Chinese households and are seen as high-priced products. The meta-analysis includes dummy variables to control the difference among food/nutrient groups. 

#### 2.2.2. Region-Level Differences

Relatively large differences appear in the magnitude of the income elasticities among regions in China. Data for rural food (0.755) and calorie (0.242) demand indicate that the mean value of income elasticity is larger than urban values (0.560, 0.144). The income elasticity of other nutrients for rural food is lower than for urban. As households meet their first-order caloric needs, they move from cheap calorie-dense staples to more expensive nutrient-dense foods [43]. The reasons for this phenomenon are divided into two aspects. First, urban households have access to a wider variety of food products, and in contrast, rural households have fewer varieties of food because their diets are dominated by households produce, and the local food markets are also underdeveloped [18]. Second, physical activity in urban households is relatively lower than the rural; thus, rural people rely on more calorie requirements, leading to diets comprising more calorie-dense foods [44]. We include a dummy variable for rural data to distinguish between urban and rural areas.

#### 2.2.3. Other Data Differences

To control the difference in data, we set up three dummy variables: (1) how the “income” (total household expenditure or total household income) is measured; (2) whether the data are micro-level survey data or macro summary data; and (3) whether the data are panel data, cross-section, or time-series data.

The primary studies include different definitions of “income” elasticity. The definition of “income” elasticity is the number of demand changes concerning changes in total income (income elasticity) or total household expenditure (expenditure elasticity). (We assume that total household income equals total household expenditures in the long run. However, in the short term, income and expenditure will vary due to savings and borrowing.) There appear to be differences between these two types of studies. For the sake of simplicity, we do not distinguish between income elasticity and total-expenditure elasticity here, while we control for this difference using a dummy variable [41]. In addition, we exclude conditional elasticities calculated by a food-demand model with a function of total food expenditure, or specific food category expenditure, because food expenditure is a family decision variable. If the studies treat total food expenditure or specific food category expenditure as exogenous, concerns arise regarding endogeneity bias [45]. Our sample includes 922 estimated food–income elasticities, and the rest (615 estimates) are total expenditure elasticities. Meanwhile, observations (119) and (113) for calorie/nutrient are total income elasticities, and the rest are expenditure elasticities. The results in Table 1 show that the mean income elasticity of food and nutrients is lower than expenditure elasticity.

Data differences may also lead to systematic differences in elasticity values [12]. Household survey data are generally considered superior to aggregate data; survey data are more consistent with demand theory and may include demographic characteristics, making it possible to examine the heterogeneity of different household preferences [46]. Furthermore, panel data are generally considered superior to cross-sectional data in controlling for unobserved heterogeneity in consumer choice [38,47]. The mean elasticities using panel data are lower than those using other data types (Table 1); thus, we control this data difference analysis using a dummy variable.

#### 2.2.4. Per Capita Income

Low income impacts the quantity and composition of food demand. According to Engel’s Law, as income increases, the proportion of income spent on food decreases. Low-income groups spend nearly half of their budget on food, while high-income groups allocate a small proportion of their income to food. Therefore, low-income groups are more responsive to food prices and income volatility, especially for high-value products [48]. To control for income effects in our meta-analysis, we include the logarithm of per capita income. (If per capita income is reported in the sample literature, we include it. If no income or total expenditure values are reported, we use per capita income values with region in the study year from the National Bureau of Statistics of China to represent it. For studies using panel and time-series data, we take the average of these incomes for the period of the data in the underlying study.)

In addition, Bennett’s Law states that as incomes rise, families change the allocation of their food budgets from starchy staples, cheap sources of calories, to more expensive foods, such as nutrient-rich fruits and animal products. Shifting consumption toward more diversified foods leads to greater nutrient content [49]. Changes in dietary behavior as a function of income may be captured by nonlinear specifications of household food products and nutrient demand functions [23]. The interaction term between the logarithm of per capita income and the different food (or nutrients) is included to explore possible differences.

#### 2.2.5. Modeling and Estimation Differences

We include dummy variables to control four types of models and estimate differences, which are as follows: (1) the type of budget process (multi-stage or single-stage); (2) the type of demand system in the sample study (or only a single equation); (3) whether the demand model is a three-rank Quadratic Almost Ideal Demand System (indicating that the Engel curve in the demand system is nonlinear); and (4) the type of the estimation program in the primary study.

Multi-stage budgeting refers to the sequential allocation of the consumer’s total spending. For example, in a two-stage budgeting model, the consumer decides on the total expenditure of food to be spent in the first stage and then decides on the amount of individual food in the second stage. Multi-stage budgeting requires a weakly separable utility function of consumers among groups of goods [50]; the restriction may affect the estimated value. As seen from Table 1, most studies used single-stage budgeting, and the mean income elasticity of food is 0.721, higher than the mean income elasticity of multi-stage studies (0.540). In contrast, the average value of multi-stage nutrient elasticity was higher than that of a single-stage budget.

Lewbel [51] classifies them among demand systems according to their rank. Demand systems applied nowadays usually have a rank of two or greater to better reflect consumers’ food consumption preference and explain food consumption in responses from different income groups. As Table 1 indicates, most of the meta-samples are two-rank models. For food consumption, the mean elasticity value of the three-rank model is higher than that of the two-rank model. The lack of sample size does not allow comparing the mean value of nutrient elasticity.

Different estimation procedures may have a connection with the estimated elasticities. The Least Squares (LS) is the most popular method, which is used in nearly half of our meta-sample, followed by the method of Quasi-Unrelated Regression (SUR). Other commonly used estimation methods include Maximum Likelihood Estimation (MLE), panel data estimation, and instrumental variables. Among the estimation methods, the mean elastic value estimated by the LS method is lower than that estimated by other methods.

#### 2.2.6. Publication Bias

Publication bias is the phenomenon of easy publication of studies with compelling empirical results of a particular magnitude or statistical significance. Such bias may be caused by several factors, including a tendency for authors, reviewers, and editors to avoid reporting and publishing small and insignificant estimates for statistically significant results. The quality of data and research may also vary among the publication types. The attributes considered include the type of publication used (i.e., published in journals/international organization reports/conference reports/working papers, etc.) and the language of the publication (i.e., published in Chinese/English). 

## 3. Method

Meta-analysis is the quantitative alternative to qualitative reviews of the empirical literature [52]. It is useful in identifying study-specific characteristics that may influence the reported results. There are three main procedures for applying meta-analysis: (i) Funnel Asymmetry Testing (FAT) to test whether publication selection bias influences the sample of estimates; (ii) Precision Effect Testing (PET) to test for a genuine non-zero effect of estimates once the publication bias is accommodated and corrected; and (iii) meta-regression analysis (MRA) to investigate whether study characteristics affected the size of the demand–income relation for food and nutrients.

### 3.1. Funnel Asymmetry Test (FAT) and Precision Effect Test (PET)

Meta-analysis is useful for identifying publication bias and estimating genuine empirical effects beyond publication bias in the effect size. Publication bias may overweigh the real estimated elasticities in the literature, causing the skewed distribution of reported effect size [52,53]. Therefore, it is crucial to identify whether the literature on a given topic suffers from publication selection bias and, if it does, how such bias should be corrected.

A common approach to test the existence of publication bias in the economics literature is the FAT approach [54], which can be specified as follows:(1)$$\;effect_{ij}=\alpha +\beta_{1}SE_{ij}+\mu_{ij}\;\;\;\;\;\;\;\;\;\;\;\;\;\;\;\;\;\;\;\;$$
where $effect_{ij}$ is the standard effect size from the primary studies, such as the reported elasticity coefficient (i.e., the income elasticity of food/calorie/nutrition) in the primary studies, i denotes the elasticity estimate for the j-th study. $SE_{ij}$ is the standard error in primary studies (in the absence of an estimated standard error, the inverse of the square root of the sample size or the inverse of the square root of the degree of freedom can also be used as a measure of estimation [55]), and $\;\beta_{1}\;$ is statistically significant when the study effect size ($effect_{ij}$) is more likely to correlate with its standard error ($SE_{ij}$). It means publication bias varies with standard error ($SE_{ij}$); large numbers of studies with lower SE values are associated with the significance of $\beta_{1}$, suggesting publication bias.

Stanley [56] proposed a genuine empirical effect beyond the publication bias. Stantely [56,57] suggested applying PET. Based on Equation (1), we consider two sets of covariates: $X_{k}$ is the vector of covariates that explains the heterogeneity associated with estimated elasticity and publication bias; $Z_{m}$ is the vector of covariates to capture the heterogeneity related to the tendency to published estimates. $\beta_{0}$, $\beta_{1}$, $\pi_{k}$, $\delta_{m}$ are estimated parameters, and $\mu_{ij}$ denotes the error term in the regression. The sign and significance of $\beta_{0}$ identify the empirical effect under review of the meta-analysis.
(2)$$effect_{ij}=\beta_{0}+\beta_{1}SE_{ij}+\sum_{k=1}^{K}\pi_{k}X_{k,ij}+\sum_{m=1}^{M}\delta_{m}Z_{m,ij}+\mu_{ij}\;$$

A common problem for the above equation is embedded heteroscedasticity due to the specification of the equation; estimates with small variance are more reliable than those with high variance. Therefore, Equation (2) cannot be estimated directly by the OLS method. Stanley [56] suggests using the inverse of the standard error to correct for heteroscedasticity. Thus, Equation (3) can be re-written as follows:(3)$$t_{ij}=\beta_{0}\left(\frac{1}{SE_{ij}}\right)+\sum_{k=1}^{K}\pi_{k}(X_{k,ij}/SE_{ij})+\beta_{1}+\sum_{m=1}^{M}\delta_{m}Z_{m,ij}+\mu_{ij}$$

Equation (3) allows us to conduct an FAT to test whether publication selection bias influences the sample of estimates. The larger the deviation between $\beta_{1}$ and 0 ($\beta_{1}\neq 0$), the greater the asymmetry of the effect size reported in the primary studies, thus suggesting publication bias. Equation (3) can also ascertain the genuine empirical effect beyond publication bias through the PET. The $\beta_{0}$ (empirical effect) implies the significance of income on elasticity. If $\beta_{0}$ is statistically significant, we conclude that the impact of income on elasticity is statistically different from 0 in the reviewed studies.

### 3.2. The MRA: Identify Sources of Heterogeneity

The MRA examines the sources of heterogeneity in the population effect size. A typical multivariate MRA is given by the following:(4)$$E_{ij}=\alpha +\sum_{k=1}^{K}\beta_{k}X_{k,ij}+\mu_{ij};\;\mu_{ij}\sim N\;(0,\;\sigma_{\mu })$$
where $E_{ij}$ is the i-th elasticity of the j-th study (including food, calorie, and nutrients), $X_{k}$ represents the vector of the determined study attribute, and $\alpha ,\;\mu_{ij}$, and $\beta_{k}$ are the estimated parameters. The signs of the reported coefficients ($\beta_{k}$) indicate how a given variable influences changes in the food/nutrient–income elasticity. Thus, a positive sign indicates that the variable positively affects elasticity, and a negative sign implies the opposite effect. The econometric procedure applied to identify the heterogeneity, particularly to reduce publication bias in MRA, is the weighted average of measures using effect size precision (for example, the inverse of the variance of the standard error of the estimated effect size). The standard error in estimating effect size is discounted for small sample studies. Given this, we adopt the method Stanley [55] suggested to estimate Equation (3)’s parameters. This method is a weighted least square model that reduces publication bias in the MRA by using the reciprocal of the square root of the standard error for estimating the weight of the major studies. The variance of each estimate is used as the weight to minimize the variance of the weighted average result.

In light of differences in the primary studies, this article analyzes study characteristics to uncover factors affecting the estimated income elasticities of food, calories, and nutrients. As for the nature of estimate elasticities examined in the primary studies, we grouped three major categories: food-, calorie- and nutrient-income elasticity (the relationship between demand for fats, proteins, minerals, and income) as dependent variables in a series of meta-analyses. The independent variables were stratified based on five dimensions: demand specification, nature of data, estimation technique, published features, and study area. The variables adopted in our study are listed and described in Table 2.

## 4. Results

### 4.1. MST-MRA Results

The initial step for meta-analysis is the FAT and PET, which are obtained through Equation (2). The results presented in Table 3 show that all the coefficients, except nutrient, are statistically different from zero, suggesting that publication bias is only investigated in the food and calorie elasticities used in the analysis. A negative bias occurs in food-income elasticity ($\beta_{1}=-31.39$, *p* < 0.001), indicating that the effect of income on food demand is negatively skewed. In addition, a negative coefficient of $SE_{ij}$ indicates that positive estimates of food–income elasticities appear to be under-reported in the sampled studies. Calorie shows a positive bias $(\beta_{1}=46.62$, *p* < 0.001), indicating the calorie elasticity value favors the positive estimated effect size. In such cases, authors do not report all the results they uncover. Instead, they select results consistent with prior findings or results they believe have a better chance of publication. $\beta_{0}$ reflects the genuine underlying empirical effect after correcting for any publication selection bias. This coefficient is also positive and statistically significant, indicating income’s positive and statistically significant effect on calorie and nutrient intake from the reviewed studies. The Z-variables in Table 3 represent the probability of journal editors’ acceptance of articles for publication. Regarding food elasticity research, the coefficients of journal variable and panel data are positive and significantly different from 0, indicating that articles published in journals and studies using panel data are more likely to exhibit publication bias. Research on calories using micro-survey data is less likely to have publication bias, and single-stage budgeting methods are more prone to publication bias.

### 4.2. Meta-Regression Analyses of Food-Income Elasticities

Table 4 presents the results for all foodstuffs, pooled using Equation (4), and estimated by weighted least squares (WLS) and OLS. Both are similar in the signs and significance levels of the estimated coefficient, indicating that our results are robust. The adjusted R^2^ estimated by WLS is higher than the result of OLS because the WLS was used to reduce the weight of outliers and improve the fitting degree of the model; this is consistent with the assumption that heteroscedasticity exists in the linear meta-regression.

The results show that studies focusing on rural populations estimated significantly higher income elasticities than those for urban or national (rural and urban jointly) populations. This is in line with our expected assumption that urban residents have access to more varieties of foodstuffs. Furthermore, our results on the effect of data attributes on the elasticity were significantly different from those used in primary studies, with lower food demand elasticities for panel data and micro-level survey data. Similar to other studies [27,29], the marginal effect for the total income dummy is negative for food (−0.166). This result may be related to the definition of total income/total expenditures; total income equals total expenditure plus net savings. If the saving rate increases, the estimated elasticities regarding total income may be lower than elasticities for total expenditure [30]. The estimated coefficient for the single-stage dummy is 0.761, indicating the baseline model with multi-rank significantly estimates higher elasticities than single rank.

There is a statistically significant relationship between food elasticities and the log of per capita income, staple food dummy, the aquatic dummy, and the interaction term of per capita income. It confirms differences in income elasticities according to the food group. Elasticities for meat and aquatic products are significantly higher than demand for other food groups; i.e., demand for these foods is most responsive to income changes. Demand for basic foods, such as staple food, is less elastic and thus less responsive to income changes. Concerning the country’s overall income level, our results show that a doubling of per capita income is predicted to lead to a decline of 0.19 in food–income elasticities (−0.27 ×  ln (2) ≈ −0.19). This is consistent with the idea of a saturation point for food consumption and reducing the share of food expenditure as income increases [4]. For staple food, the total marginal effect, including the interaction term, is −0.322–0.270=−0.592, so a doubling of per capita income would lead to a decline of  ln(2)×0.592 ≈ 0.41 in the income elasticity for staple food. With the economic growth, people’s demand for staple foods will become less income-sensitive, indicating that Chinese has met energy needs from starchy staples [6]. For aquatic products, the product of significant public health concern in China, the corresponding increase in income elasticities for aquatic products is about 0.27. Optimizing the consumption structure of animal food and increasing the aquatic products may reduce the risk of heart disease and promote brain and eye health [58]. The increased cost of a healthy diet coincided with higher incomes, so more and more people may seek high-priced aquatic products.

The results show a negative, albeit statistically insignificant, coefficient for the variable representing articles published in journals. Additionally, most of the coefficients for the estimation methods (Model_type, Budget_stage) are not statistically significant. It seems that the estimation procedure and type do not influence much in terms of estimated income elasticities.

### 4.3. Meta-Regression Analyses of Calorie and Nutrient–Income Elasticities

Table 5 presents the results for the calorie and nutrient meta-regression. It explains how publication bias distorts the elasticities—peer-reviewed journals have significantly higher nutrient-income elasticities than working papers/reports. In addition, the marginal effect for Chinese publications is −1.546, which is statistically significant, indicating income elasticity for nutrients in Chinese-language publications is lower than those in English-language publications. The type of data used in primary studies confirms significant differences in nutrient- and calorie-income elasticities. The panel and time-series data appear to have significantly higher estimates for nutrients. However, the calorie–income elasticities of studies that employed panel time-series data were significantly lower in magnitude than those using cross-sectional data. Regarding the functional form of the demand model, the only statistically significant variable is using a demand system for nutrients. Compared with the pragmatic model, the estimated coefficient for the demand system dummy is −1.187, implying that the demand system model tends to yield lower income elasticities for nutrients. As for the multi-stage budgeting model, the only statistically significant result is positive for calories. This result can be related to the fact that a multi-stage budgeting assumption restricts consumption flexibility to adjust to income changes [39].

The nutrient–income elasticity estimated using total income is significantly smaller than the elasticity of total expenditure. There is a significantly negative relationship between income growth and calorie–income elasticity, while the magnitude of the decline in income elasticities is small (about 0.09 in response to a doubling of per capita income,  ln(2)×−0.121 ≈ 0.09). For the income elasticity of nutrient products, the vitamins dummy and the interaction term between the log of per capita income and the vitamins dummy are statistically significant. The income elasticities for nutrients (as a whole), fat, and minerals do not change significantly with income growth. However, for vitamins, the total marginal effect, including the interaction term, is 0.169−0.0122=0.157, so a doubling of per capita income would lead to an increase of ln(2)×0.157 ≈ 0.11 in the income elasticity for vitamins. It appears that the intake of vitamins will exhibit greater decreases in the face of economic downturns than other nutrients. In particular, results here suggest that as countries become richer, not only are calorie intakes response to income change on a gentle trajectory, but that the impact on nutrient intake is likely to be small.

## 5. Discussion and Conclusions

This study aims to better understand the relationship between income level and the demand for different foods, nutrients, and calories in China. This will help in understanding what domestic policies can be used in the fight against malnutrition. A significant contribution of the study is creating a database of food-income elasticities that can identify the factors underlying differences in estimates in China. A meta-sample of income elasticities for China was built, drawn from 64 primary studies, covering 1537 food-income elasticities for eight groups of food (Staple food, Vegetables and fruit, Meat, Oil and fat, Dairy, Aquatic products, Eggs, and Other food), 153 nutrient–income elasticities for three types of nutrients (fat, vitamin, and minerals), and 147 calorie–income elasticity estimates. The variables capturing a set of study-specific attributes identified important factors associated with the variation in primary studies to explain the heterogeneity across income elasticities.

On average, income elasticities in China are positive across all food categories. Thus, income is a potential key determinant of food demand. According to the meta-analysis results, elasticities for meat and aquatic products are significantly higher than demand for other food, while staple food is less elastic. Moreover, there is a significant negative relationship between income growth and the size of food–income elasticities, that is, the food–income elasticities decline with income growth. This relation also holds for staple food. One exception is the aquatic products; we find a positive relationship between income growth and the magnitude of income elasticity of aquatic products. A possible explanation is that the saturation point of staple food consumption has been reached due to increased incomes. Chinese people may shift consumption toward more diversified and often more expensive foods with higher nutrient content [59]. These results suggest that income growth may align with a beneficial shift in the nutritional status of diets to reduce the risk of heart and brain disease [58].

Concerning nutrients income elasticities, we found the mean income elasticity is higher for fat (0.324), vitamin (0.304), and protein (0.303) than for minerals (0.248) and calories (0.212). The impact of income on calorie and nutrient intake is likely to be small, as some studies have concluded [38,60]. Knowing how calorie and nutrient elasticities change with income becomes necessary in light of nutrition-related chronic diseases. There is a significantly negative relationship between income growth and calorie–income elasticity size, while the predicted decline is relatively small (0.09 in response to a doubling in per capita income). However, given that China’s per capita income is growing by 7.2%, the results suggest that the sensitivity of caloric demand to income will be constant though incomes rising. Thus, for countries that are already consuming well beyond the recommended calorie levels, further increases in income will lead to an even larger consumption of calories. Thus, policy-makers still should primarily focus on the alleviation of weight-related problems, with one in six Chinese adults falling into obesity [6,8]. As for nutrients, only the income elasticity for vitamins may have a gentle increase with income growth, while the impact on other nutrients is not significant. The effect of nutrients on the chronic disease has also been documented. Vitamin D supplementation has been associated with reduced mortality. Zinc deficiency has caused children growth retardation. Vitamin A, Vitamin E and total antioxidant capacity supplementation have had a protective effect on metabolic syndrome [61]. Regarding the steadily increasing incidence of nutrition-related chronic diseases [1], economic growth accompanied by nutrition transition patterns tends to evoke health problems. Some studies suggest that good nutrition is a driver of economic growth. Therefore, development policies should be geared specifically toward reducing chronic malnutrition to spur economic growth rather than focusing on economic growth to spur good nutrition [62,63]. Policy-makers should continue to monitor the evolution of demand for these nutrients to ensure people’s health, particularly given the sheer size of the population and the relatively tight nutrition situation in China.

Our results prove the relationships between regions and food-, calorie-, and nutrient-income elasticity. Regarding food, we found that rural people are more sensitive to income and price change, which is consistent with earlier findings in the literature [63,64]. We did not find significant effects of urban and rural differences concerning calories and nutrients.

As for the role of data and methodology, we found that the food elasticity is higher when the primary studies substitute household expenditures for income, similar to [29]. In addition, we found publication bias in studies on food and calorie elasticities; the elasticity values reported in journals were lower than those in reports or working papers, which is in line with the previous studies [27,28]. Specifically, we found that income elasticity for nutrients in Chinese-language publications tends to be lower than in English-language publications.

Overall, the results suggest that development strategies to improve economic growth may improve diet quality but may be insufficient to improve nutrient intake. This study’s findings also suggest that heterogeneity in food-, calories-, and nutrients-income demand elasticities in China are mainly due to contextual characteristics and methodological factors. Notably, we also find publication bias. It is crucial to be aware of these causes of heterogeneity and provide reliable income elasticity estimates to improve food demand projections and design effective food and nutrition policies in China. This study’s main limitation is that the number of examined articles may be insufficient; only a few studies investigate nutrient–income elasticity in China, much less than food demand. Due to the nature of our study, relying on a more extensive study set does not seem feasible for the current meta-analysis. Understanding the impact of income changes on nutrient intake is still an important direction worthy of further research.

## Figures and Tables

**Figure 1 nutrients-14-04711-f001:**
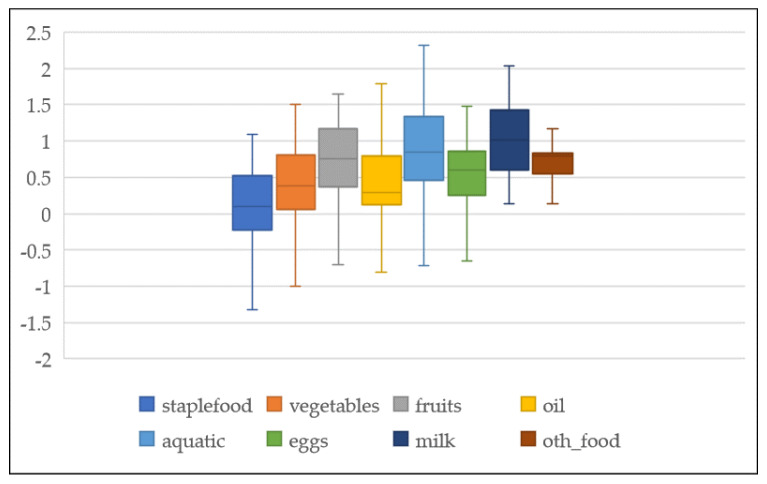
Box plot of food–income elasticities synthesis.

**Figure 2 nutrients-14-04711-f002:**
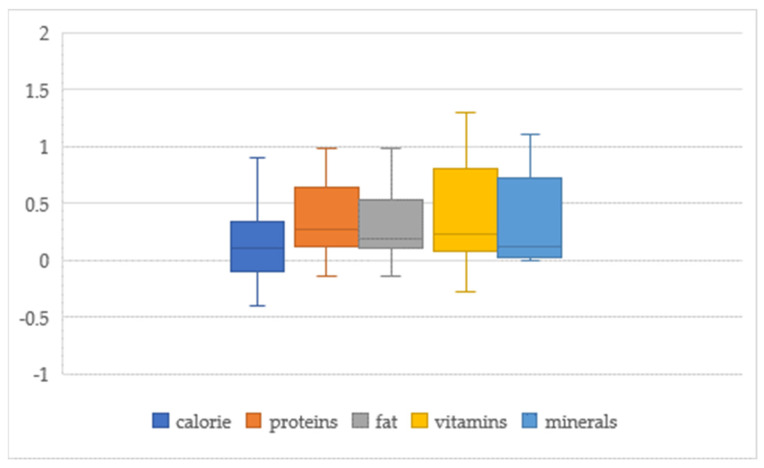
Box plot of calorie-, and nutrient-income elasticities synthesis.

**Figure 3 nutrients-14-04711-f003:**
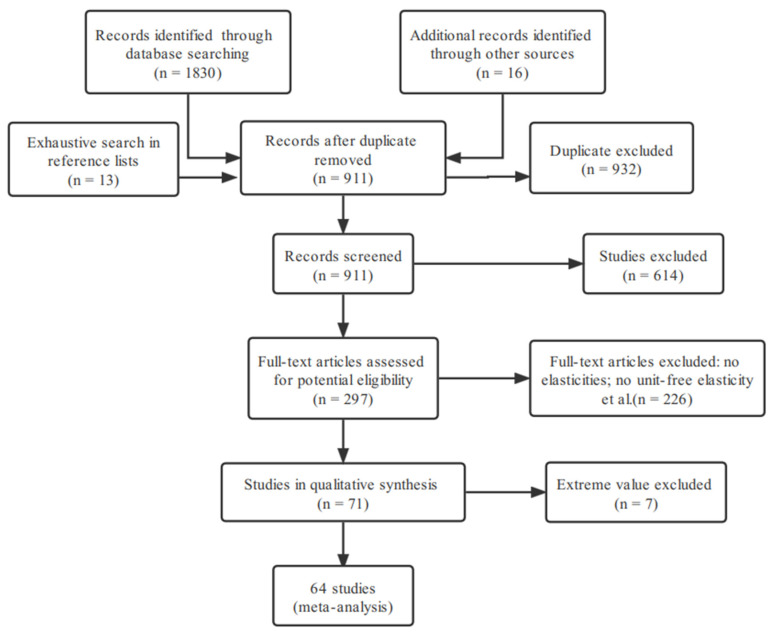
Selection of primary studies and construction of meta-sample.

**Table 1 nutrients-14-04711-t001:** Summary statistics of meta-sample.

Income Elasticities	Food-Income Elasticities			Calorie-Income Elasticities			Nutrient-Income Elasticities		
	Number	Mean	Std. Dev.	Number	Mean	Std. Dev.	Number	Mean	Std. Dev.
** *Total* **	1537	0.690	0.685	147	0.212	0.325	153	0.298	0.315
** *Published features* **									
English	385	0.724	0.362	70	0.187	0.284	69	0.249	0.290
Chinese	1152	0.678	0.766	77	0.233	0.367	84	0.342	0.345
Journal	1291	0.695	0.703	105	0.203	0.342	107	0.333	0.334
Other	246	0.662	0.593	42	0.288	0.293	46	0.281	0.304
** *Area* **									
Rural	676	0.755	0.579	49	0.242	0.267	46	0.225	0.184
Urban/Nation	861	0.560	0.615	56	0.144	0.377	61	0.317	0.340
** *Data* **									
Macro-aggregate	553	0.722	0.804	63	0.178	0.269	61	0.270	0.229
Micro-survey	984	0.654	0.568	84	0.265	0.368	92	0.350	0.371
Panel	953	0.439	0.672	70	0.137	0.181	77	0.166	0.165
Time series	96	0.567	0.447	17	0.206	0.104	23	0.244	0.029
Other data	488	0.680	0.491	60	0.324	0.351	54	0.526	0.193
** *Income measure* **									
Expenditure	615	0.924	0.376	28	0.194	0.149	23	0.386	0.384
Income	922	0.531	0.796	119	0.216	0.350	130	0.288	0.318
** *Type of estimator* **									
FE/RE	104	0.878	0.744	14	0.100	0.001	23	0.042	0.030
IV/GMM	64	0.923	0.087	28	0.527	0.215	31	0.467	0.257
LS	973	0.493	0.608	84	0.113	0.280	84	0.205	0.275
MLE	64	0.937	0.221	7	0.985	n/a	8	0.903	n/a
SUR	332	0.908	0.488						
** *Type of model* **									
Demand system	1351	0.777	0.659	70	0.280	0.362	77	0.387	0.392
Single equation	186	0.059	0.544	77	0.157	0.297	77	0.249	0.219
** *Type of budget* **									
Multi	264	0.540	0.686	28	0.346	0.379	23	0.601	0.544
Single	1273	0.721	0.685	119	0.179	0.314	130	0.263	0.283
Model_rank 3	326	1.062	0.482		n/a	n/a	8	0.114	0.000
Model_rank 2	1025	0.590	0.699	147	0.227	0.325	145	0.317	0.321
** *Type of food* **									
Staple food	242	0.146	0.807	-	-	-	-	-	-
Vegetables and fruit	245	0.434	1.096	-	-	-	-	-	-
Meat	270	0.865	1.099	-	-	-	-	-	-
Oil and fat	151	0.492	0.673	-	-	-	-	-	-
Aquatic products	215	1.066	1.078	-	-	-	-	-	-
Eggs	220	0.744	1.894	-	-	-	-	-	-
Dairy	91	1.084	0.613	-	-	-	-	-	-
Other food	105	1.102	0.759	-	-	-	-	-	-
** *Type of nutrients* **									
Protein	-	-	-	-	-	-	20	0.303	0.296
Fat	-	-	-	-	-	-	16	0.324	0.326
Vitamin	-	-	-	-	-	-	8	0.304	0.235
Minerals	-	-	-	-	-	-	6	0.248	0.426

Note: n/a: not applicable. Std. Dev.: Standard Deviation. FE/RE: Fixed Effects/Random Effects. IV/GMM: Instrumental variables estimation/Generalized Methods of Moments. LS: Least Squares. MLE: Maximum Likelihood Estimate. SUR: Quasi-Unrelated Regression.

**Table 2 nutrients-14-04711-t002:** Description of the variables.

Category	Variables	Description
Published features	Pub_journal	Dummy variable: 1 = peer-reviewed journal, 0 = report/working paper
	Pub_chinese	Dummy variable: 1 = Chinese, 0 = English
Study area	H_region	Dummy variable: 1 = rural, 0 = other (including urban and nation)
Income measure	H_income	Dummy variable: 1 = total income, 0 = total expenditure
Data	D_micro	Dummy variable: 1 = micro-level survey data, 0 = macro-level aggregate data
	D_panel	Dummy variable: 1 = panel, 0 = others
	D_time series	Dummy variable: 1 = time series, 0 = others
Model and method	Model_type Budget_stage	Dummy variable: 1 = demand system, 0 = pragmatic model Dummy variable: 1 = single-stage, 0 = multi-stage
	Model_rank	Dummy variable: 1 = model_rank 3, 0 = model_ rank 2
Per capita income level	lncome	Continuous variable: Log of per-capita annual disposable income
Food group	Staple food, Vegetables and fruit, Meat, Oil and fat, Dairy, Aquatic products, Eggs, Other food	Dummy variable: 1 = * food, 0 = others
Nutrient group	Protein, Fat, Vitamin, Minerals	Dummy variable: 1 = * nutrient, 0 = others
Interaction *	* lnincome	Interactions between individual food or nutrient dummy variables (represented by *) and logarithms of per capita income

Note: Individual food or nutrient dummy variables (represented by *)

**Table 3 nutrients-14-04711-t003:** The results of FAT-PET.

Variables	Food	Calorie	Nutrition
1/SE (empirical effect-$\beta_{0}$)	0.384	0.417 *	0.661 *
	(1.64)	(1.89)	(1.85)
X-variables			
H_income/SE	−0.116	−0.230	−0.406
	(−1.05)	(−0.99)	(−1.18)
Model_type/SE	0.767 ***	−0.240	−0.284
	(3.15)	(−1.50)	(−0.89)
Model_rank/SE	0.0695		
	(0.83)		
Z−variables			
Pub_journal	22.55 ***	−7.183	0.327
	(4.90)	(−0.64)	(0.02)
H_region	4.195	−8.175	−6.106
	(1.34)	(−0.70)	(−0.40)
D_micro	1.942	−21.98 *	−11.42
	(0.86)	(−1.97)	(−0.40)
D_panel	4.690 *	−11.53	−13.29
D_time series	(1.99)	(−0.90)	(−0.75)
	−3.443	−27.91	−2.424
	(−0.90)	(−1.64)	(−0.10)
Budget_stage	0.532	32.28 **	17.26
	(0.16)	(2.57)	(0.90)
Constant (publication bias−$\beta_{1}$)	−31.39 ***	46.62 ***	33.91
	(−5.78)	(3.59)	(1.25)
Number of observations	1537	147	153
R^2^	0.997	0.821	0.577

Note: *t*-values are given in parenthesis. * *t* < 0.1, ** *t* < 0.05, *** *t* < 0.01.

**Table 4 nutrients-14-04711-t004:** Results from the meta-regressions of food–income elasticities.

	Variables	OLS	WLS
Publication	Pub_journal (1 = peer-reviewed journal, 0 = report/working paper)	−0.0567	−0.0793
		(−0.21)	(−0.41)
	Pub_chinese (1 = Chinese, 0 = English)	−0.129	0.0570
		(−0.57)	(0.37)
Study area	H_region(1 = rural, 0 = other)	0.311 *	0.745 ***
		(1.98)	(5.15)
Data	D_micro (1 = survey data, 0 = aggregate data)	−0.296	−0.312 **
		(−1.36)	(−2.11)
	D_panel (1 = panel, 0 = others)	−0.109	−0.214 *
		(−0.69)	(−1.75)
	D_time series (1 = time series, 0 = others)	−0.219	0.559
		(−0.48)	(0.45)
Income measure	H_income (1 = total income, 0 = total expenditure)	−0.354 **	−0.166 *
		(−2.03)	(−1.69)
Model and method	Model_type (1 = demand system, 0 = pragmatic model)	0.185	0.382
		(0.69)	(1.38)
	Budget_stage (1 = single-stage, 0 = multi-stage e)	−0.0515	−0.0788
		(−0.25)	(−0.51)
	Model_rank (1 = model_rank 3, 0 = model_ rank 2)	0.490 **	0.761 ***
		(2.20)	(4.17)
Types of food	Staple food	−1.833	−1.422 *
		(−0.47)	(−1.98)
	Vegetables and fruit	4.185	5.473
		(1.65)	(1.59)
	Oil and fat	−1.783	−0.252
		(−0.91)	(−0.13)
	Dairy	0.701	−2.107
		(0.23)	(−0.86)
	Aquatic products	−2.179	3.009 ***
		(−1.35)	(−2.99)
	Eggs	−1.540	0.883
		(−0.89)	(0.63)
	Other food	−1.085	−3.526 *
		(−0.58)	(−1.76)
Income	lnincome	−0.206	−0.270 *
		(−0.69)	(−1.40)
Interaction *	Staple food * lnincome	−0.231	−0.322 *
		(0.53)	(1.95)
	Vegetables & fruit * lnincome	−0.484	−0.552
		(−1.67)	(−1.48)
	Oil and fat * lnincome	0.239	0.0574
		(1.06)	(0.25)
	Dairy product * lnincome	−0.100	0.217
		(−0.30)	(0.79)
	Aquatic products * lnincome	0.240	0.359 ***
		(1.23)	(2.80)
	Eggs * lnincome	0.231	−0.0395
		(1.09)	(−0.22)
	Other food * lnincome	0.148	0.414 *
		(0.68)	(1.84)
	Constant	2.348	5.371
		(0.60)	(1.23)
	Number of observations	1537	1516
	Number of studies	58	57
	R^2^	0.540	0.746

Note: *t*-values are given in parenthesis. * *t* < 0.1, ** *t* < 0.05, *** *t* < 0.01.

**Table 5 nutrients-14-04711-t005:** Results from the meta-regressions of calorie-, nutrient–income elasticities.

	Variables	Calorie	Nutrient
Publication	Pub_journal	0.107	−1.546 *
		(0.50)	(−11.48)
	Pub_chinese	−0.0428	−0.711 **
		(−0.26)	(−21.56)
Study area	H_region	0.128	0.0553
		(0.87)	(3.45)
Data	D_micro	−0.0791	0.171
		(−0.33)	(2.46)
	D_panel	−0.268	1.012 **
	D_time series	(−0.69)	(17.45)
		−0.946 *	1.183 **
		(−1.86)	(13.58)
Income measure	Income	−0.247	1.176 **
		(−0.99)	(−10.64)
Model and method	Model_type	−0.478	−1.187 **
		(−1.34)	(−35.57)
	Budget_stage	0.844 ***	0.623
		(3.73)	(6.14)
Types of nutrients	Fat		1.526
			(5.26)
	Vitamin		−1.216 *
			(11.57)
	Minerals		6.596
			(3.25)
Income	lnincome	−0.121 *	−0.0122
		(1.95)	(0.60)
* lnincome	Fat * lnincome		−0.0796
			(−2.36)
	Vitamin * lnincome		0.169 **
			(−54.04)
	Minerals * lnincome		−0.757
			(−3.58)
	Constant		2.261 **
			(15.00)
	Number of observations	147	153
	Number of studies	20	19
	R^2^	0.942	0.989

Note: *t*-values are given in parenthesis. * *t* < 0.1, ** *t* < 0.05, *** *t* < 0.01.

## Data Availability

The dataset used during the current study are available from the corresponding author on reasonable request.

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
