# Peer review of "The Income Elasticities of Food, Calories, and Nutrients in China: A Meta-Analysis"

_nutrients, 2022, doi:10.3390/nu14224711_

Round 1
Reviewer 1 Report
The manuscript entitled “The income elasticities of food, calories, and nutrients in China: A meta-analysis” is the study with an aim to review all publication in China that examines the estimates for income elasticity of food, calorie, and nutrients.
Introduction – “hidden hunger” the term “malnutrition” is more commonly used and can explain both problem of undernutrition and obesity due the lack of certain nutrients in diet
- Report on Chronic Disease (reference is missing)
- “summary in Section 2.2.” - whether it refers to table 1 or to the entire subchapter
considering that “elasticities” is economic value, it would be good to give some introductory definition of the term
e.g. “Income and price elasticities of food demand are economic measurements of the responsiveness of food consumption to income and price changes for a group of consumers. The income elasticity of food demand measures the percent change in the consumption of total food or a certain food item or group of food items to a percent change in the real income of consumers (DOI : https://doi.org/10.2499/p15738coll2.134675)“
…and for example, you have unnecessary definitions such as what is meta-analysis which are given as school definitions
Also, in the introduction you should mention what is the situation outside the China
Description of data
- - what is the reason for mixing scientific papers and various reports (you call it grey)
- - in the manuscript is not defined how this additional 13 papers entered in the analysis.
- You state that SEVERAL records were excluded due the effect size is not available and in scheme is shown that almost 70% are excluded because there is no full-text article?!?!
- - the full abbreviation name is required (for NBSC)
- basic knowledge in the field of nutrition is necessary for interpreting data connected with food - everything that contributes to energy intake (all macronutrients) contributes to calories - you cannot mix foods and nutrients
- also statement that rural-households consume more calorie dense-food, and les nutrient dense-food is in conflict with the sentences that precede that conclusion (where you stated that in rural households’ people are more relayed on home-cooked, and urban on pre-prepared foods?!)
Results
- Table 4 missing abbreviation explanation
- the results need to be better explained in the way of public health
-
Discussion
- What do you mean under 7 types of food
- Again, you cannot mix foods and nutrients
- necessary to write more clearly
References should be prepared according to instructions
Reviewer 2 Report
The manuscript is interesting and provides relevant information on the income elasticity of food, energy and nutrients in Chile. In this regard, the information presented by the authors will allow analysis of critical aspects of food and nutrition in the Chinese population. The structure of the manuscript is good. The methodology used is sufficient and allows confirming the hypothesis of the study. However, I do have some comments.
I. Minor Comments:
1. Improve the wording of the objective of the manuscript
2. It would be important to briefly discuss aspects related to nutrient deficiencies (especially micronutrients) and health.
3. I suggest including a figure that summarizes the main findings of the study.
Round 2
Reviewer 1 Report
I would like to thank the authors for their efforts to improve the manuscript.
Some rearrangements and clarifications have been made to the text. The paper is now somewhat clearer.
According to the grey literature it is truth that lot of important data can be lost if we do not include this kind of literature, but as it is already stated in Schmucker et. al (2017) “even the most comprehensive search for gray and unpublished data will not allow a final judgment whether the identified sample is in fact complete and representative for all of the hidden data”.
Some changes should be done
e.g.
line 111/112 In the selection process, the reference list in previous review studies of food/nutrient demand was… - it turned out very vague - they cannot be called “previous studies” when it was about the current research
line 158 - The main food groups - within the science of nutrition it is known what the major food groups are so please change the terminology at least to „main food grouping in this research”
The proofreading by a native speaker is suggested.
